# Ladderpath Approach: How Tinkering and Reuse Increase Complexity and Information

**DOI:** 10.3390/e24081082

**Published:** 2022-08-05

**Authors:** Yu Liu, Zengru Di, Philip Gerlee

**Affiliations:** 1International Academic Center of Complex Systems, Beijing Normal University, Zhuhai 519087, China; 2Department of Mathematical Sciences, Chalmers University of Technology, 405 30 Gothenburg, Sweden; 3Department of Mathematical Sciences, University of Gothenburg, 405 30 Gothenburg, Sweden

**Keywords:** complexity, syntactic information, Shannon entropy, Kolmogorov complexity, hierarchy, module, origin of life, alien signal, language, evolution

## Abstract

The notion of information and complexity are important concepts in many scientific fields such as molecular biology, evolutionary theory and exobiology. Many measures of these quantities are either difficult to compute, rely on the statistical notion of information, or can only be applied to strings. Based on assembly theory, we propose the notion of a *ladderpath*, which describes how an object can be decomposed into hierarchical structures using repetitive elements. From the ladderpath, two measures naturally emerge: the ladderpath-index and the order-index, which represent two axes of complexity. We show how the ladderpath approach can be applied to both strings and spatial patterns and argue that all systems that undergo evolution can be described as ladderpaths. Further, we discuss possible applications to human language and the origin of life. The ladderpath approach provides an alternative characterization of the information that is contained in a single object (or a system) and could aid in our understanding of evolving systems and the origin of life in particular.

## 1. Introduction

In an interview, John Maynard Smith noted that while the 19th century had been the century of energy, in which science and engineering were concerned with transforming energy from one form to another (chemical to mechanical as in a steam engine or mechanical to electrical as in a dynamo), the 20th century was about information, and in particular the transformation of information [1]. This shift has been particularly pronounced in biology where the discovery of DNA as the carrier of hereditary information, and the subsequent sequencing of the entire human genome has shaped biology into a science of information.

Although most of us have an intuitive idea of what constitutes information, defining the term rigorously turned out to be challenging. This is partly due to the discrepancy in the meanings of the word *information*. In one sense information refers to knowledge or facts about the world (i.e., semantic information), whereas it can also refer to a characterization of the object structure that is independent of any interpretation or meaning (i.e., syntactic information) [2].

Consider the sentence: *Claude was an American mathematician, and Claude spent his childhood in Michigan.* The semantic information of this sentence is impossible to compute without properly defining the constituent words. We need to understand what “mathematician” refers to in the real world, which cannot be deduced from this sentence on its own. Therefore when considering an object in isolation the only sensible measure of information is a syntactic one.

When considering the transmission of signals along the communication channel, which was Claude Shannon’s motivation for formulating the information theory, we would like each transmitted symbol to be as informative as possible. However, when we turn our eye to the human language or to biology, we note that information is often conveyed by structures that contain redundancy.

These structures, which we often think of as complex, are somewhere between completely ordered (e.g., a crystal) and completely random (e.g., a perfect gas). From the intuitive point of view we would not like to assign a high complexity to a highly ordered sequence (e.g., a sentence with repetitive words), but on the other hand, we would not like to call a totally random sequence as complex since it lacks any structure. Instead, we would like to assign large complexity to sequences somewhere in between the repetitive and the random [3,4]. This requirement, motivated by our intuition, makes the definition of complexity challenging.

The first attempt to quantify complexity along these lines was made by Kolmogorov [5] and later refined by Chaitin [6,7,8], and is referred to as algorithmic complexity. It defines the complexity of a sequence as the length of the shortest possible computer program that could generate the sequence. This sounds promising, but due to the halting problem of the universal Turing machine, it has the practical drawback that there is no universal method for calculating the complexity of an arbitrary string [9,10]. Notwithstanding, algorithmic complexity could be approximated when defined on a weaker class of Turing machines (i.e., in particular contents or environments). For example, the well-known Lempel-Ziv complexity and algorithm that is widely used for file and data compression [11,12,13], related to an optimal rate of lossless string compression. Another example is the physical complexity that is particularly devised to characterize the amount of information about the surrounding environment that has been encoded in genomic sequences [14].

Another family of complexity measures build on Claude Shannon’s information theoretic concepts and in particular on Shannon entropy, which is defined as $H=-\sum p_{i}\log_{2}p_{i}$, where the sum runs over all possible symbols in the sequence and $p_{i}$ is the probability of observing symbol *i* (Shannon entropy itself is not a useful measure of complexity since it is maximized for completely random sequences) [15]. For example, Grassberger introduced a measure known as Effective Measure Complexity which is defined as the average amount of information contained in a string that can be used when guessing the next symbol [16]. A similar measure termed Statistical Complexity was introduced by Crutchfield and Young which measures the Shannon entropy of the probability distribution of the causal states present in the sequence [3]. In statistics, another related concept is Fisher information, representing the curvature of the relative entropy of the distribution of a random variable with respect to its parameters, which measures the information this random variable carries. It has a wide range of applications including measuring the complexity of learning tasks [17] and analyzing signals generated by the market [18].

Unlike the above-mentioned algorithmic complexity which is a form of absolute information of the individual object [19] and thus sensitive to the particular pattern of the sequence, the Shannon-entropy-related complexity measures rely on a statistical notion of information, which is insensitive to sequences. They can only be defined for sequences (or other objects) drawn from a statistical ensemble or equivalently sequences that are infinitely long. A solution to this problem is to assume that the underlying probability distribution for a specific sequence is uniform across all characters that appear in the sequence. However, this approach ignores the information (or “meaning”) stored in the particular sequence. The question of how to characterize the syntactic information content and the complexity of finite length single instantiations thus remains open.

Since information is related to repeated patterns (as the Lempel–Ziv algorithm, for example, which utilizes repetitive substrings to compress information [20,21]), the above questions can also be approached from another angle, in a more general sense. The idea of joining or alternating objects that have already existed to construct new objects (e.g., gene sequences, molecules, technological inventions) was conceived in 1977 by François Jacob, phrased as “evolution as tinkering” [22]. This idea of accumulating information has laid the basis for many research fields such as protein interactions [23,24] and software modularity [25,26], and continues to inspire new research from a more general perspective, e.g., the evolution of network complexity where reuse plays a significant role [27]. In 1997 Donald Knuth extensively studied the “addition chain” in his famous computer science book, which has a similar motivation: The shortest addition chain for *n* can be considered the most efficient addition sequence to reconstruct an integer from integers that have been constructed previously [28]. More recently, the concepts of “pathway complexity” [29] and “assembly space” [30,31] were developed to characterize the object complexity, by counting the number of construction steps, in which already built structures can be reused in subsequent constructing processes. This theory has been applied to detect biosignatures, and the hypothesis is that if a sufficiently complex molecule is found in abundance, biotic processes must have been involved [32]. Lately, the concept of “molecular assembly tree” was developed to characterize the hierarchical relationships within a group of distinct molecules, which shows great potential in fields such as drug discovery and origin of life research [33].

Following those lines above, i.e., “evolution as tinkering”, “addition chain”, and especially “assembly theory”, in this paper we formalize an alternative approach to characterize information and complexity by focusing on the process of generating given objects, e.g., sequences, sentences, pictures, molecules, proteins, architectural structures. We call this the *ladderpath approach*. It is grounded not in the abstract theory of computation such as Kolmogorov complexity or Shannon entropy, but rather, as suggested by assembly theory, inspired by the tinkering process and reuse in construction that has shaped the living world.

In the following Section 2 we elaborate our *ladderpath approach* from scratch by giving related definitions one by one, followed by introducing an algorithm to calculate the ladderpath of an object. In Section 3 we relate the ladderpath to evolution, and lastly in Section 4 we discuss our findings in the context of interpreting unknown signals and ideas about life/evolution.

## 2. Ladderpath Approach: Starting from Scratch

### 2.1. A Thought Experiment

In order to motivate the need for the definition of a ladderpath let us consider the following thought experiment: From somewhere in the universe, we have received a sequence of letters (assuming we have somehow translated it into Latin letters):

ACXLGICXGOXEMZBRCNKXACXLPICXEMZBRCNKX.

Does this string contain any information, if yes, how could we understand or interpret it? First of all, we may say that this string contains information in the semantic sense, but we abandon this idea because we have no means of assigning a meaning to the individual letters in the string.

How do we approach the problem of extracting information from the string? We may start by searching for repetitions in the sequence. In the string, EMZBRCNKX appears twice. However, in such a short string, the probability that this 9-letter substring appears twice from randomness alone is very low (if it appears 3, 4, 100 or more times, we may be more certain that it must not appear randomly, and must have a specific semantic meaning). Although appearing twice is not completely impossible, here we assume that anything that repeats twice or more is not random. Therefore we assume that EMZBRCNKX must have a specific semantic meaning. Nonetheless, by just looking at this string itself, we cannot infer what EMZBRCNKX really refers to; If we frequently see EMZBRCNKX appears together with other texts, pictures, objects, etc., we have the possibility to infer what it refers. However, for now, we can only say that EMZBRCNKX must mean something or it must be a word or phrase in an alien language. In addition, we see ACX appears twice so it also means something.

How about other non-repetitive substrings such as LGIC and LPIC? Do they have particular meaning or semantic information? From this single string we can never figure it out. However, if we find LPIC repetitively appears in some other places or sources, we can be sure that it must have a particular meaning. Thus, for the original string viewed in isolation, information in the semantic sense is just contained in EMZBRCNKX and ACX. (In fact, this string is “*we live in Jupiter. we love Jupiter*”. We just converted a letter into another (including space and period): EMZBRCNKX is “*jupiter*”. (notice period and space), and ACX is “*we.*”, while LPIC is *love*, and LGIC is *live*).

### 2.2. Definition of *Generation-Operation*

Let us now move one step closer to a rigorous definition of a ladderpath. The motivation comes from the following question: Imagine we have a set that includes letters A, B, C, D, E, and F, and we need to obtain a specific string
X=ABCDBCDBCDCDEFEF
then what is the quickest way (i.e., the least number of steps needed) to obtain X? Surely, this question can be extended to other situations, e.g., we have a set of atoms and need to obtain a specific molecule (a preliminary idea in the case of molecules has been investigated in our previous publication [33]); or we have a set of hardware and need to obtain a bicycle; or we have a set of ingredients and condiments, and need to obtain a specific dish. Here we just take the string as a simple but generic example.

First of all, we define a ***basic set***, denoted as $S_{0}=\{$A(0), B(0), C(0), D(0), E(0), F(0)} which is defined to be a special type of *partially ordered multiset*, i.e., its elements are partially ordered (the ordered parts are separated by double slash “⫽”, between which we can call “one ***level***”; and the unordered parts are separated by comma “,”), and the number of instances (called *multiplicity*) of each element is written in the brackets behind the element. Note that the basic set $S_{0}$ should be predefined, according to the specific research problem currently at hand, after which the analysis can be conducted (see Section 4.1 for more discussions). In this specific case we defined the basic set to be constituted from single letters.

Any element in this type of partially ordered multiset is called a ***building block*** (block for short). The elements in the basic set are called ***basic (building) blocks***. In addition, the string we want to obtain, X, is called the ***target (building) block***. Note that, in $S_{0}$, as shown above, the basic blocks are all unordered and their multiplicities are all zeros. Then, we define an operation on this type of partially ordered multiset:***Generation-operation*** is defined as: take any number of blocks in the partially ordered multiset (the multiplicity decreases accordingly, but note that any block can be taken even if its multiplicity is zero or negative) and combine them in a certain way (note that the newly-generated blocks must not be present in the set), and then put the combined one back into the set at the level that is one level higher than the highest level of the constituted blocks. After this operation, a new partially ordered multiset of this type is obtained.
Note that: (1) It differs from the joining process in assembly theory [29,33] in that it allows any number of blocks to be joined and the putting-back step keeps the partial order. (2) The “negative multiplicity” is only a transient notation used for simplicity purposes, which does not appear in the final result. (3) Here “a certain way” could be different for different systems. For example, for the string system here, we define “a certain way” as: write the strings in the same line from left to right; For molecules, there could be a range of different generation-operations that could help capture properties like molecular isomerism (referring to Section 4.3 for more discussions). (4) “at the level that is one level higher than the highest level of the constituted blocks” means: for example, if the newly-generated block is made from blocks at level 1, level 3, and level 5, respectively, then the newly-generated block must be put at level 6.

For example, the operation of taking A and C in $S_{0}$, combining them into AC, and then putting AC back is a generation-operation on $S_{0}$, which can be denoted as $S_{0}$:A + C = AC $\to S^{\prime }$ = {A(−1), B(0), C(−1), D(0), E(0), F(0) ⫽ AC(1)}. Note that AC is put back at the next level of A and C, indicated by the symbol “⫽”.

For another example, the operation of taking D, D, F, AC in $S^{\prime }$, combining them into DDFAC and then putting it back is a generation-operation on $S^{\prime }$, denoted as $S^{\prime }$:D + D + F + AC = DDFAC $\to S^{\prime \prime }$ = {A(−1), B(0), C(−1), D(−2), E(0), F(−1) ⫽ AC(0) ⫽ DDFAC(1)}. Because AC is at the highest level among D, F and AC which is level 2, so DDFAC is put back at level 3.

For a third example, the operation of taking B, C in $S^{\prime \prime }$, combining them into BC and then putting it back is a generation-operation on $S^{\prime \prime }$, denoted as $S^{\prime \prime }$:B + C = BC $\to S^{\prime \prime \prime }$ = {A(−1), B(−1), C(−2), D(−2), E(0), F(−1) ⫽ AC(0), BC(1) ⫽ DDFAC(1)}. Note that BC is put back at level 2, because B and C are at level 1.

Now, let us get back to the original question: how to obtain the target block X. The following example (Ex1) shows several successive generation-operations after which we obtain X:


Ex1.1st generation-operation, $S_{0}$:C + D = CD $\to S_{1}$ = {A(0), B(0), C(−1), D(−1), E(0), F(0) ⫽ CD(1)}      .2nd, $S_{1}$:B + CD = BCD $\to S_{2}$ = {A(0), B(−1), C(−1), D(−1), E(0), F(0) ⫽ CD(0) ⫽ BCD(1)}      .3rd, $S_{2}$:E + F = EF $\to S_{3}$ = {A(0), B(−1), C(−1), D(−1), E(−1), F(−1) ⫽ CD(0), EF(1) ⫽ BCD(1)}      .4th, $S_{3}$:A + BCD + BCD + BCD + CD + EF + EF = X$\;\;\;\;\;\;\;\;\;\to S_{4}$ = {A(−1), B(−1), C(−1), D(−1), E(−1), F(−1) ⫽ CD(−1), EF(−1) ⫽ BCD(−2) ⫽X(1)}      .Lastly, take one X out from $S_{4}$, and then we achieve our goal: obtained one target block X. The last step can be considered one special generation-operation, that is, only take out but do not put back, which can be denoted as $S_{4}$:X(−1) $\to S_{5}$ = {A(−1), B(−1), C(−1), D(−1), E(−1), F(−1)⫽CD(−1), EF(−1) ⫽ BCD(−2) ⫽X(0)}


From $S_{0}$ and $S_{1}$ alone, we can infer that the generation-operation must be C + D = CD. In general, the same argument applies to $S_{i}$ and $S_{i+1}$. Therefore, we could simply write this path through which we achieve our goal as $S_{0}\to S_{1}\to S_{2}\to S_{3}\to S_{4}\to S_{5}\Rightarrow X$. Evidently, there are many paths through which we can achieve the same goal. Here is another sequence of operations that also result in X,

Ex2.1st generation-operation, $S_{0}$:D + B + C = DBC $\to S_{1}^{\prime }$ = {A(0), B(−1), C(−1), D(−1), E(0), F(0) ⫽ DBC(1)}      .2nd, $S_{1}^{\prime }$:E + F = EF $\to S_{2}^{\prime }$ = {A(0), B(−1), C(−1), D(−1), E(−1), F(−1) ⫽ DBC(1), EF(1)}      .3rd, $S_{2}^{\prime }$:A + B + C = ABC $\to S_{3}^{\prime }$ = {A(−1), B(−2), C(−2), D(−1), E(−1), F(−1) ⫽ DBC(1), EF(1), ABC(1)}      .4th, $S_{3}^{\prime }$:DBC + DBC = DBCDBC $\to S_{4}^{\prime }$ = {A(−1), B(−2), C(−2), D(−1), E(−1), F(−1) ⫽ DBC(−1), EF(1), ABC(1) ⫽ DBCDBC(1)}      .5th, $S_{4}^{\prime }$:C + D = CD $\to S_{5}^{\prime }$ = {A(−1), B(−2), C(−3), D(−2), E(−1), F(−1) ⫽ DBC(−1), EF(1), ABC(1), CD(1) ⫽ DBCDBC(1)}      .6th, $S_{5}^{\prime }$:ABC + DBCDBC + D + CD + EF + EF = X$\;\;\;\;\;\;\;\;\;\to S_{6}^{\prime }$ = {A(−1), B(−2), C(−3), D(−3), E(−1), F(−1) ⫽ DBC(−1), EF(−1), ABC(0), CD(0) ⫽ DBCDBC(0) ⫽X(1)}      .Lastly, $S_{6}^{\prime }$:X(−1)$\;\;\;\;\;\;\;\;\;\to S_{7}^{\prime }$ = {A(−1), B(−2), C(−3), D(−3), E(−1), F(−1) ⫽ DBC(−1), EF(−1), ABC(0), CD(0) ⫽ DBCDBC(0) ⫽X(0)}

This path can be denoted as $S_{0}\to S_{1}^{\prime }\to S_{2}^{\prime }\to S_{3}^{\prime }\to S_{4}^{\prime }\to S_{5}^{\prime }\to S_{6}^{\prime }\to S_{7}^{\prime }\Rightarrow X$.

### 2.3. Definition of *Ladderpath*

For a path $S_{0}\to S_{1}\to S_{2}\to \cdots \to S_{n}\Rightarrow X$ through which we obtain the target block X, we construct another partially ordered multiset in the following way: Take the final set $S_{n}$, delete all of the blocks with zero multiplicity, and then set all other multiplicities to be the absolute value of the corresponding multiplicities (with the partial orders preserved). This procedure generates a new partially ordered multiset *J*, which we call the ***ladderpath*** of X that corresponds to this particular path.Any block in the ladderpath is called a ***ladderon***.

So, the ladderpath corresponding to Ex1 is:(1)$$J_{X,1}=\left\{A,B,C,D,E,F⫽\mathrm{CD},\mathrm{EF}⫽\mathrm{BCD}\left(2\right)\right\}$$

Note that we often omit the multiplicity “(1)” for simplicity. Likewise, the ladderpath corresponding to Ex2 is:(2)$$J_{X,2}=\left\{A,B\left(2\right),C\left(3\right),D\left(3\right),E,F⫽\mathrm{DBC},\mathrm{EF}\right\}$$

Evidently, every path through which the target block is obtained corresponds to one ladderpath, but not vice versa. For example,

Ex3.1st generation-operation, $S_{0}$:D + B + C = DBC $\to S_{1}^{\prime }$ = {A(0), B(−1), C(−1), D(−1), E(0), F(0) ⫽ DBC(1)}      .2nd, $S_{1}^{\prime }$:E + F = EF $\to S_{2}^{\prime }$ = {A(0), B(−1), C(−1), D(−1), E(−1), F(−1) ⫽ DBC(1), EF(1)}      .3rd, $S_{2}^{\prime }$:A + B + C + DBC + DBC + D + C + D + EF + EF = X$\;\;\;\;\;\;\;\;\;\to S_{3}^{\prime \prime }$ = {A(−1), B(−2), C(−3), D(−3), E(−1), F(−1) ⫽ DBC(−1), EF(−1) ⫽X(1)}      .Lastly, $S_{3}^{\prime \prime }$:X(−1) $\to S_{4}^{\prime \prime }$ = {A(−1), B(−2), C(−3), D(−3), E(−1), F(−1) ⫽ DBC(−1), EF(−1) ⫽X(0)}

This path can be denoted as $S_{0}\to S_{1}^{\prime }\to S_{2}^{\prime }\to S_{3}^{\prime \prime }\to S_{4}^{\prime \prime }\Rightarrow X$, and we can see that it also corresponds to ladderpath $J_{X,2}$. For a better understanding of the definition of the ladderpath, refer to Appendix B for more examples of the ladderpath of strings

It is worth mentioning that

Any target sequence has at least one ***trivial ladderpath*** in which all the ladderons are basic blocks.

Taking X as an example, its trivial ladderpath is $J_{X,0}$ = {A, B(3), C(4), D(4), E(2), F(2)}, which represents many paths that only use the basic blocks in any order to make X.

The significance of defining a ladderpath lies in the fact that one ladderpath represents many different but equivalent paths that generate the target block, meaning that the ladderpath completely filters out all of the tedious information of the steps along paths. The information retained in a ladderpath (including ladderons and their multiplicities, and the order relationships among ladderons) completely but non-redundantly describes all of the equivalent paths. The ladderpath has several good properties:One ladderon is a block that has been reused (i.e., been part in a generation-operation at least twice). The reuse is the ultimate reason why we can simplify the process of making the target block (later we shall see that this type of simplification is closely related to the “information” contained in the target block).For any ladderpath $J_{X}$,
(3)$$N_{a}=\sum_{i\in J_{X}}\left(m_{i}\times n_{i,a}\right)$$
holds for every letter, where $N_{a}$ is the number of times the letter *a* appears in the target sequence X, *i* represents all ladderons in this ladderpath $J_{X}$, $m_{i}$ is the ladderon *i*’s multiplicity, and $n_{i,a}$ is the number of times the letter *a* appears in the ladderon *i*.Take $J_{X,1}$ as an example. The letter B appears 3 times in the target sequence X, i.e., $N_{B}=3$. On the right hand side of Equation (Equation 3), the contribution of ladderon BCD is 2×1=2 (as its multiplicity is 2 and the letter B appears once in BCD); the contribution of ladderon B is 1×1=1 (as its multiplicity is 1 and the letter B appears once in B); and the contribution of other ladderons is zero. So the right hand side is also 2+1=3. It is straightforward to see that Equation (Equation 3) holds for all other letters A, C, D, E and F, too. Likewise, we can also verify ladderpath $J_{X,2}$ and $J_{X,0}$.

Any ladderpath can be fully described by a partially ordered multiset (namely, the partially ordered multiset representation, e.g., Equations (Equation 1) and (Equation 2)); Yet sometimes it is more intuitive to represent a ladderpath by a graph or network. We denote such a graph a ***laddergraph***. An example is shown in Figure 1a, where the levels represent the order relationships in the ladderpath, and the links among blocks represent generation-operations (e.g., E and F link up to EF, representing that EF is generated from E and F).

To make the laddergraph more concise, we make several conventions:1.Dim all the lines that are linked to the basic blocks.2.If a block *i* at a lower level can be linked to a block *j* at a higher level via other blocks, then even if *i* and *j* are directly linked, we do not draw the lines between *i* and *j*. For example, in Figure 1a, the block CD at a lower level is directly linked to the target block X at the higher lower since in the ladderpath $J_{X,1}$, CD is directly involved when generating X; but because CD is linked to BCD, and BCD is linked to X, then we should not draw a line between CD and X.3.The multiplicities of blocks can either be written down explicitly (as in Figure 1) or not.

In principle, the partially ordered multiset representation and the laddergraph representation of a ladderpath are one-to-one, yet because of the convention (2) and (3) above, the laddergraph representation may lose some information. Nevertheless, the most important information of the ladderpath is well-retained in the laddergraph, i.e., the hierarchical relationships among ladderons.

### 2.4. Definition of *Ladderpath-Index* (λ), *Order-Index* (ω), and *Size-Index* (*S*)

In this subsection, we will introduce three significant concepts: the ladderpath-index, and its associated concepts, size-index and order-index. However, before that, we need to first define the “length unit of a ladderpath” and the “length of a ladderpath”.

For the string examples this paper describes, we define the ***length unit of a ladderpath*** as follows: for each operation, that concatenates any two strings (namely, blocks), we associate a unit called a “***lift***” (note that since how building blocks are combined together might be different in different systems, the length unit of a ladderpath could be defined differently, yet we always term the unit as a “*lift*”).

Since each generation-operation actually corresponds to writing *n* blocks together, equivalent to repeating the action of “writing two blocks together” (n−1) times. So, every generation-operation actually corresponds to a (n−1)
*lifts*. For example, in the path in Ex1 that $S_{0}\to S_{1}\to S_{2}\to S_{3}\to S_{4}\to S_{5}\Rightarrow X$, the length of the 1st generation-operation C + D = CD is 1 *lift*, the length of the 2nd generation-operation B + CD = BCD is 1 *lift*, the length of the 3rd E + F = EF is 1 *lift*, and the 4th A + BCD + BCD + BCD + CD + EF + EF = X is 6 *lifts*. For the last special generation-operation, we can consider it as writing X and another hypothetical empty symbol together, and its length is thus 1 *lifts*. For anther example, in the path in Ex2, the length of each generation-operation is 2,1,2,1,1,5,1
*lifts*, respectively. In the path in Ex3, the length of each generation-operation is 2,1,9,1 *lifts*, respectively.

As mentioned, the length unit of a ladderpath might be defined differently. For example, when studying chemical molecules, the length unit may be defined to be the formation of one chemical bond between atoms, ions, or molecular fragments; when studying evolution, the length unit may be defined to be the emergence of a new species, or a new physiological function; and so on. Therefore, the length unit of a ladderpath is a user-defined quantity. Nevertheless, as long as it is well-defined beforehand, it is not allowed to be altered in the following analyses.

We can now make the following definition:The ***length of a ladderpath*** *J*, denoted as |J|, is the sum of the lengths of all generation-operations along the ladderpath *J*. Note that, the length of any path is thus naturally defined as the sum of the lengths of all generation-operations along this path.

|J| is to describe the “cost” that is required to generate the target block, associated with this ladderpath *J*. One ladderpath often corresponds to many paths that can generate the target block, but the lengths of all of these paths are identical (which is one of the convenient properties of a ladderpath). This implies that we can pick any of such paths to calculate |J|. For example, it is easy to verify that the length of the paths in Ex2 and Ex3 are both 13 *lifts*, and both paths indeed correspond to one ladderpath $J_{X,2}$ (evidently, we can conclude that $|J_{X,2}|=13$
*lifts*). For the path in Ex1, its length is 10 *lifts*, so the length of its corresponding ladderpath $J_{X,1}$ is also 10 *lifts* (which is shorter than $J_{X,2}$).

We now define the “ladderpath-index”:The ***ladderpath-index*** of target block X, denoted λ(X), is the length of the ***shortest ladderpath(s)*** of X (there may be one or several shortest ladderpaths). Thus, the length unit of ladderpath-index is also “*lift*”.

Note that (1) ladderpath-index differs from assembly index [31,33] in that the latter is measured by counting the joining steps while the former is measured by the length of generation-operations where the length unit must be predefined for different systems; (2) Computing the shortest ladderpaths is not trivial at all. Although we have proved that it is at least as hard as an NP-complete problem (i.e., this problem cannot be solved in a polynomial time scale as the size of the problem increases), we were still able to develop a rigid procedure and algorithm, referring to Section 2.7 for details.

For the target block X, $J_{X,1}$ is its shortest ladderpath indeed. So, X’s ladderpath-index is 10 *lifts*, i.e., λ(X)=10
*lifts*. In fact, for any target block X, its ladderpath-index λ(X) describes one of its intrinsic properties, i.e., the smallest “cost” to generate X. The shortest ladderpaths is a central concept since it correspond to the most compressed way of storing the information contained in X, as we shall see soon.

We now proceed to the second critical concept, the “size-index”:The ***size-index*** S(i) of any block *i* (e.g., the target block or a ladderon) is the length of its shortest trivial ladderpath (there could be one or several shortest trivial ladderpaths, but they all have the same length).

If the basic blocks are single letters, the size-index is the number of letters in the string, e.g., size-index of X is 16 *lifts*, because its trivial ladderpath corresponds to the one where we generate X by just concatenating single letters.

The reason why the definition emphasizes the word “shortest” is that if the basic set includes not only single letters but also strings, the lengths of trivial ladderpaths might be different. For example, let the basic set be {A, B, C, D, E, F, BCD} for X. The trivial ladderpath could then correspond to concatenating single letters as before, i.e., {A, B(3), C(4), D(4), E(2), F(2)} which has the length of 16 *lifts*. Alternatively, we could employ multi-letter strings in the basic set and obtain a trivial ladderpath of the form {A, BCD(3), C, D, E(2), F(2)} whose length is 10 *lifts*. In this case, we must choose the length of the shortest trivial ladderpath to be its size-index.

We now introduce the concept of “order-index”:The ***order-index*** ω(X) of the target block X is defined to be:
(4)ω(X):=S(X)−λ(X),
where S(X) is the size-index of X, and λ(X) is the ladderpath-index of X. Evidently, the unit of order-index is “*lift*”, too.

This means that the order-index is equivalent to the number of *lift*s that are saved when generating the target block X via the shortest ladderpath compared to the trivial ladderpath, or in other words, the work that has been saved from combining blocks when constructing the target.

Last, we introduce an important property of the length of a ladderpath, which is crucial for calculating ladderpath-index and order-index:The length of a ladderpath $J_{X}$ can be directly calculated from its partially ordered multiset representation, with no need to convert the ladderpath into one of its corresponding path. The length can be calculated as:
(5)$$|J_{X}|=S\left(X\right)-\sum_{i\in J_{X}}m_{i}\cdot \left(S\left(i\right)-1\right),$$
where S(X) is the target block X’s size-index, S(i) is the ladderon *i*’s size-index, $m_{i}$ is the ladderon *i*’s multiplicity, and *i* represents all ladderons in this ladderpath $J_{X}$.

For example, in Ex1 the ladderpath was given by $J_{X,1}$ = {A, B, C, D, E, F ⫽ CD, EF ⫽ BCD(2)}. X’s size-index is 16 *lifts*; the size-index of any basic block is evidently 1 *lift* (so their contributions are always 0, and thus no need to consider); the size-indices of CD, EF, BCD are 2,2,3
*lifts*, respectively; so, $|J_{X,1}|=16-\left(2-1\right)-\left(2-1\right)-\left(3-1\right)\times 2=10$
*lifts*. Likewise, $|J_{X,2}|=16-\left(3-1\right)-\left(2-1\right)=13$ *lifts*. As we can see, the lengths calculated in this method are identical with the ones calculated directly through the definition.

In fact, based on Equation (Equation 5) and the definition of the order-index, the order-index of the target block X can be readily calculated from the following formula:
(6)$$\omega \left(X\right)=\sum_{i\in J_{X}}m_{i}\cdot \left(S\left(i\right)-1\right),$$
where *i* represents all ladderons in this shortest ladderpaths.

### 2.5. Ladderpath-Index and Order-Index Are Two Axes of “Complexity”

The shortest ladderpaths of the target block X contains “the whole knowledge” about how to describe its complexity and the information it carries; The ladderpath-index and order-index calculated from the shortest ladderpaths are abstractions of this whole knowledge in different aspects:The ladderpath-index λ(X) describes the amount of “information” that X carries, that is, how many extra steps/“costs” or how much extra “information” the external agent needs to input in order to generate X, equivalent to the difficulties to reproduce X (which is distinct from the “information” that Shannon entropy or thermodynamic entropy refers to);The order-index ω(X) describes how much “information” can be saved, i.e., the amount of redundant “information” (equivalently, the difference between the trivial ladderpath and the shortest ladderpath). This is consistent with the intuition, as the more steps/“costs” (namely, how many *lift*s) it saves, the more ordered the target block X is;Now we can see that “complexity” that we often intuitively talk about has two aspects: One is described by the ladderpath-index λ which focuses more on the difficulties and costs of constructing the target; While the other aspect is described by the order-index ω which focuses more on how the target is built in an organized and hierarchical manner;Lastly, for a particular target (or target system), the sum of its ladderpath-index and its order-index is always equal to its size-index, which automatically solves the “normalization” problem that may occur when comparing different sized targets.

To be more clear, we consider a simple example: consider the sequence in Section 2 which had some structure, X = ABCDBCDBCDCDEFEF, and another random sequence with the same length, W = ABCDEFCFEDCBFDBA. Intuitively, the information that W carries is more than that carried by the non-random X, because, to repeat W, the amount of information needed (i.e., needed to be memorized) is larger than that needed to repeat X. On the other hand, the random W is less ordered/organized compared to X. These two aspects are intuitively consistent with the two concepts λ and ω: On one hand, λ(W)=16>λ(X)=10, i.e., the information W carries is more than that X carries; on the other hand, ω(W)=0<ω(X)=6, i.e., W is less ordered/organized than X.

The concepts we have introduced apply not only to strings, but also to any other objects. Based on the definition of the order-index Equation (Equation 4), we can draw Figure 2a, where each diagonal is the contour line of the size-index *S*. We can see that under the same ladderpath-index (i.e., contains the same amount of information and is equally difficult), the more ordered an object is, the larger it is; while under the same size-index, the more ordered the object is, the smaller the ladderpath-index is, i.e., the less information it contains.

Now, we take a detailed example to further explain how the concept of ladderpath describes the “complexity” of objects. Consider the patterns shown in Figure 2b, which can be interpreted as stones placed in a clearing. These patterns can be described using the ladderpaths shown in Figure 2c (see Appendix C for details on how to compute the ladderpaths of these stone patterns), and from these the corresponding ladderpath-index and order-index can be computed.

Based on these we make the following observations:1.For patterns with the same size-index (e.g., [i], [iii] and [vi]), as ω increases, the pattern becomes more and more ordered, which is consistent with our intuitions. On the other hand, as the ladderpath-index λ increases, the pattern becomes more and more difficult to reproduce (e.g., to reproduce [vi], we need to memorize the position of almost every stone, but to reproduce [i], we only need to memorize the description of the ladderpath, rather than the position of each stone), which is also consistent with our intuitions;2.There are some counter-intuitive points indeed, yet they are also the most important: For example, [i] is more ordered than [ii], but the difficulty to generate either of them is the same, as their λ’s are identical. [i] is more ordered than [iii] and [vi], but it is less difficult and takes fewer *lift*s to generate [i]. [i] is more ordered than [v], yet not only is it less difficult to generate [i], the size of [i] is also larger than [v] (the same argument applies to [ii], compared with [iv], [v] and [vi]; and so on). We shall see later that this point is the very key to explaining why the emergence of life is not as difficult as imagined before.

It deserves to mention that we may intuitively say that [vi] is more “complex” than [i], because the former looks more random/irregular/difficult to reproduce; while we may also say that [i] is more “complex” than [vi], because [i] needs more detailed, complex and delicate mechanisms to generate. However, it is not difficult to realize that these two “complex” refer to two distinct directions. The former is about how much information the system contains (or equivalently, how difficult, how much external information we need to input, to reproduce the system); while the latter is about how organized the system is. In fact, the two directions correspond to the ladderpath-index λ and the order-index ω, respectively. Therefore, now we are able to distinguish the two axes of “complexity”.

The final remark on the two axes of complexity is that we always have λ(X)+ω(X)≡S(X), indicating that the two axes are not independent for a particular target or target system X. It is true indeed, but in fact, any target can be placed in a particular position in these coordinates. This is because although the three indices are constrained by λ+ω≡S, two of them are free. Referring to Figure 2, if we fix the size-index of a target, by rearranging the patterns of this target, it can move freely in the coordinates (e.g., imaging rearranging among pattern [i], [iii] and [vi]). It is exactly because of the internal structure and information of this target, its coordinates are fixed. Note that λ and *S*, or ω and *S* could be chosen as the axes, but our motivation here is to relate the intuition of complexity with the axes, and the intuition of complexity often comes from difficulty and order, as we have discussed above. To emphasize, the size is a significant factor of the complexity of an object, but it is not simply a proportional relationship—not simply the larger the size, the more complex—for example, referring to the Figure 2b, pattern [i] is larger than [v] but [i] is easier to be reproduced (as the ladderpath-index of [i] is smaller). This is exactly why we need the indices λ and ω to represent the two axes of complexity. With these concepts, we can now readily compare the complexity of objects with different sizes.

### 2.6. Extension of the Concept: The Ladderpath of a Whole System

Up until now, we have discussed the ladderpath of a single target block, but all of these concepts can be extended to a whole system of blocks. To do that, we need to consider all of the blocks in the target as a whole and as one huge “target block”. For example, if in the system, there are 2 strings of AAB, 3 strings of BC and 1 string of DDF, we can then consider them as one huge “target block”: the string “AAB, AAB, BC, BC, BC, DDF”, which we term the ***target system*** for convenience (but note that these strings are not connected, which is distinct from the single string AABAABBCBCBCDDF). It deserves to mention that the ladderpath of a target system is different from the concept of the “molecular assembly tree” which only handles a group of distinct types of molecules [33].

Now we take an example to illustrate the ladderpath of a target system (the extension of other concepts is then straightforward). Imagine the target system we need to obtain in the end is:Q={ABDEDBED(2),ABDED,ABDABD,CAB(2),ED(3)}
while the basic set is $U_{0}$ = {A(0), B(0), C(0), D(0), E(0)}. So, the following successive generation-operations make one path:

Ex4.1st generation-operation, $U_{0}$:A + B = AB $\to U_{1}$ = {A(−1), B(−1), C(0), D(0), E(0) ⫽ AB(1)}      .2nd, $U_{1}$:C + AB = CAB $\to U_{2}$ = {A(−1), B(−1), C(−1), D(0), E(0) ⫽ AB(0) ⫽ CAB(1)}      .3rd, $U_{2}$:AB + D = ABD $\to U_{3}$ = {A(−1), B(−1), C(−1), D(−1), E(0) ⫽ AB(−1) ⫽ CAB(1), ABD(1)}      .4th, $U_{3}$:ABD + ABD = ABDABD $\to U_{4}$ = {A(−1), B(−1), C(−1), D(−1), E(0) ⫽ AB(−1) ⫽ CAB(1), ABD(−1) ⫽ ABDABD(1)}      .5th, $U_{4}$:E + D = ED $\to U_{5}$ = {A(−1), B(−1), C(−1), D(−2), E(−1) ⫽ AB(−1), ED(1) ⫽ CAB(1), ABD(−1) ⫽ ABDABD(1)}      .6th, $U_{5}$:ABD + ED = ABDED$\;\;\;\;\;\;\;\;\;\;\;\;\to U_{6}$ = {A(−1), B(−1), C(−1), D(−2), E(−1) ⫽ AB(−1), ED(0) ⫽ CAB(1), ABD(−2) ⫽ ABDABD(1), ABDED(1)}      .7th, $U_{6}$:ABDED + B + ED = ABDEDBED$\;\;\;\;\;\;\;\;\;\;\;\;\to U_{7}$ = {A(−1), B(−2), C(−1), D(−2), E(−1) ⫽ AB(−1), ED(−1) ⫽ CAB(1), ABD(−2) ⫽ ABDABD(1), ABDED(0) ⫽ ABDEDBED(1)}      .Lastly, take all the blocks included in the target system Q from $U_{7}$, and then the goal is achieved. The last step can be considered one special generation-operation, i.e., take out but do not put back, denoted as $U_{7}$:Q(−1) $\to U_{8}$ = {A(−1), B(−2), C(−1), D(−2), E(−1) ⫽ AB(−1), ED(−4) ⫽ CAB(−1), ABD(−2) ⫽ ABDABD(0), ABDED(−1) ⫽ ABDEDBED(−1)}
Therefore, the ladderpath that corresponds to this path above is (its laddergraph representation is shown in Figure 3):(7)$$J_{Q}=\left\{A,B\left(2\right),C,D\left(2\right),E⫽\mathrm{AB},\mathrm{ED}\left(4\right)⫽\mathrm{CAB},\mathrm{ABD}\left(2\right)⫽\mathrm{ABDED}⫽\mathrm{ABDEDBED}\right\}$$

In fact, this path is Q’s shortest ladderpath indeed (verified by our algorithm, detailed in Section 2.7). So, based on Equation (Equation 6), we are able to calculate Q’s order-index ω(Q)=1+1×4+2+2×2+4+7=22
*lifts*. As Q’s size-index is S(Q)=8×2+5+6+3×2+2×3=39
*lifts* (namely, the total number of letters included in all of the strings in Q), we can then use Equation (Equation 4) to calculate its ladderpath-index λ(Q)=39−22=17 *lifts*. We can verify this result by looking at the total length of all the generation-operations in the corresponding path: The lengths of the first six generation-operations are all 1 *lifts*, the 7th generation-operation has the length of 2 *lifts*, and the last special generation-operation has the length of 9 *lifts*, so the answer is also 17 *lifts*.

### 2.7. Algorithm to Compute the Shortest Ladderpaths

To find the shortest ladderpath(s), we first developed the procedure that can give us one ladderpath each time, regardless of its length. Here we illustrate this procedure in the case of a target system instead of a target block, as the former is a more general case (without losing generalities, taking strings as an example):1.First, create an empty multiset ℋ to store blocks in. Later the ladderpath can be readily computed from ℋ;2.Starting from the target system Q, we preserve only one instance of each type of distinct blocks in Q, and put all other repetitions into ℋ;3.Keep slicing the blocks in 𝒬 (i.e., slicing strings into multiple substrings or letters) in a pre-determined systematic manner, until in 𝒬 there are at least two substrings (or letters) that are identical. Preserve only one of such identical substrings in 𝒬 and put all other repetitions into ℋ. Note that there could be many systematic manners to slice the string, referring to Appendix D for an example;4.Repeat from step 3, until no repetitive substrings or letters can be found in Q. Then, cut all of the remaining substrings into basic blocks (i.e., single letters in this case), and put them all into ℋ. Now, ℋ records one ladderpath;5.The final step is to align the (sub)strings in ℋ in the right hierarchical order (that is, based on how they are sliced: which is the parent and which are the children), which results in the partially ordered multiset representation of this ladderpath.

See Appendix D for a detailed example. The output of the above procedure will depend on how we slice the strings. If we go through all deterministic slicing schemes we will generate all possible ladderpaths of the target system. From this collection of ladderpaths we select the shortest one. This gives us the algorithm to obtain the shortest ladderpaths. The code of this algorithm is available at https://github.com/yuernestliu/ladderpath, (accessed on 2 August 2022).

The motivation here is to show a proof-of-concept algorithm and code to calculate the shortest ladderpath for a small target or target system, rather than providing an efficient and practical algorithm to deal with long strings or sequences (which might be used for string compression or gene sequence analysis for example). The idea is that if an algorithm for short strings can be specified, other algorithms for long strings, images, molecules, proteins, 3D objects, etc could be developed and sophisticated in the future.

Indeed, as Lempel-Ziv is also a lossless compression algorithm [12,13], it shares some similarities with this ladderpath algorithm in the aspects of general ideas of hierarchy and a few coding techniques. Nevertheless, besides their different motivations, the outputs of the two algorithms, resulting from their calculated slicing scheme respectively, are also different.

The final remark on the current version algorithm is that, it is not feasible to enumerate all ladderpaths for a large system as the number of possible ways to slice a string increases exponentially with its length. In fact, finding the shortest ladderpaths is at least as hard as an NP-complete problem, because it is a more complex version of the *addition chain* problem (first introduced by Knuth in his book [28]) which has been proven to be NP-complete [34].

Nevertheless, in practice it might be acceptable to just have “short-enough” ladderpaths. Then there might be shortcuts. For example, we can slightly change the above algorithm:
In step 3, we bias the algorithm to search for repetitive substrings of maximum length.
In this way, although the ladderpath obtained is not guaranteed to be the shortest rigorously, it tends to be often short enough. The intuition is that the longer the ladderons (i.e., repetitive substrings) are, the more generation-operations are saved, and thus the shorter the resulted ladderpath is.

## 3. Ladderpath’s Significance in Evolution: *Ladderpath-Systems*

A system’s ladderpath-index λ corresponds to the least “cost” needed to generate the system. If we assume that the probability associated with each generation-operation is identical, then λ would be inversely proportional to the overall probability to generate this system, i.e., the larger λ is, the less probable the system can be eventually generated. For example, as shown in Figure 2, generating /reproducing pattern [i] is easier than generating /reproducing pattern [vi].

However, to safely reach this statement “generating [i] is easier than generating [vi]”, a key assumption has to be made. This key assumption is: Any block that has been generated in previous steps can be reused in any amount, with no need to generate it from scratch (that is to say, the number of any block that has been generated is immediately infinite as if any of such block has an infinite reservoir; or equivalently, the number of ladderons is always infinite). For example, the last ladderon of pattern [i] (referring to Figure 2c, the pattern that consists of 8 stones) is reused once (i.e., repeated twice), meaning that the length of the trivial ladderpath of the last ladderon, namely 7 *lifts*, is saved. This kind of reuse is the very reason that makes generating [i] simpler than generating [vi].

In certain real-world scenarios, this type of reuse is possible (but not in every scenario). For example, in technology, if a new invention gets known in the society, it can be reused from that time on, with no need to reinvent it, that is, the reuse of ladderons is possible. However, if one invents something but no other person knows it, the reuse of this invention would not be possible from the perspective of the whole society, meaning that the system (namely, the society) will not behave as the ladderpath theory describes.

Naturally, a further question arises: What conditions must a system satisfy such that it can be described by the ladderpath process? The answer is, we need two conditions—the first is the ability to generate new blocks, and the second is that blocks can replicate and thus increase in number. The first condition is implied in the generation-operation, while the second is implied in the assumption that the number of any ladderon is infinite. So,
we call a system as a ***ladderpath-system*** if it satisfies the two conditions: (i) the ability to generate new blocks, and (ii) some or all blocks can replicate;
Intriguingly, these two conditions hold in many scenarios/systems, such as life, language, and technology.

Notice that for the first condition “the ability to generate new blocks”, new blocks do not have to be generated by combining two existing blocks, that is, the generation-operation does not have to be exactly like the description in Section 2.2, and it can be defined differently according to the specific system in question. For example, if we consider single cells, not only gene recombination and horizontal gene transfer can be defined to be generation-operations (as the way before, i.e., by combining existing blocks), a genetic mutation can also be defined to be a generation-operation (in a different way, i.e., by changing an existing block). As for language, people create new words by combining existing words, borrowing words from other languages, and so on, resulting in new blocks generated. For technology, inventing is the way to generate new blocks which are always made by recombining or revising what already existed.

As for the second condition “replication”, refers to either self-replication, or duplication with the aid of other things. That is, if a block can be categorized as a ladderon, it must have the ability to replicate. All ladderons are blocks, but only the blocks that are able to replicate become ladderons. Which blocks can replicate and which cannot set the stage for the “natural selection”. One of the prominent properties of life is the ability to self-replicate: an individual and a species can self-replicate; and DNA can also replicate (it is actually the network consisting of DNA, RNA, and protein that can replicate [35,36,37]). For languages, the newly-created words and phrases can be used by other people is the way to replicate (when long-distance communication is not as convenient as today, the words that are newly created by local people hardly propagate in the whole society, so seeing from the whole society, these new words cannot become ladderons, which might be the mechanisms that a language is divided into different accents or evolves to different languages). For technology, if inventions are well-documented in papers, patents, etc., such that they can be reproduced, they can be considered as being able to replicate.

Now, that we have introduced the concept of ladderpath-systems, we may further address the following question: do totally random structures (white noise for example) carry the most information, or is the opposite true? There are two different angles to look at the “information” that the question refers to. (1) To repeat the particular sequence of a white noise, we need to memorize the whole sequence, which itself, in a sense, necessitates the great amount of explicit information. (2) Yet, complexity does not only mean the “difficulty to repeat” but also the amount of underlying hierarchical information that involves repetition and selection, which is another layer of information. We argue that the second layer of information is taken care of by the two properties of ladderpath-systems mentioned above. That is, the abundance of such selection-associated information is indicated by the ladderpath-index and the order-index both having a high value. Therefore, in this sense, white noise is almost devoid of such information, since only its ladderpath-index is high whereas its order-index is extremely low.

To end this subsection, the origin of life is likely to have occurred via this “ladderpath mechanism” (not in a “magical event”), that is, during the process lots of ladderons (which are able to replicate) must have been generated. In fact, there is evidence for that: e.g., obviously cells and species can self-replicate; the molecules that are made of individual cells or constitute chemical reaction networks (e.g., autocatalytic sets [35,36,37]) are indeed able to replicate.

## 4. Discussions

In this paper we have introduced the concept of a ladderpath of an object, which makes it possible to characterize the complexity of an object along two axes: the ladderpath-index and the order-index. We will now discuss how this construction plays out in two different context: ***isolated*** and ***non-isolated /united*** systems.

### 4.1. On Information: Alien Signals (Isolated System)

Let us imagine we have received a sequence Y of letters from another planet (some signals which have been converted to a sequence of letters):Y=TBCDEFRBCDEFTEFHKREFHJKLMUVTEFPSMU
and we need to figure out if it contains any kind of information. First of all, Y can be considered as an ***isolated system***, because there are no other signals or sequences that can be considered together with Y.

By employing the ladderpath theory and the algorithm we have developed (Section 2.7), we find Y’s shortest ladderpath
$$J_{Y}=\left\{T,B,C,D,E,F,R\left(2\right),H\left(2\right),K\left(2\right),J,L,V,T,P,S,M,U⫽\;\mathrm{EF}\left(2\right),\mathrm{MU}⫽\mathrm{TEF},\mathrm{BCDEF}\right\}$$
and its ladderpath-index is λ(Y)=34−1×2−1−2−4=25
*lifts*. As a result, we can interpret Y as T [BCD [EF]] R [BCD [EF]] [T [EF]] H K R [EF] H J K L [MU] V [T [EF]] P S [MU], where the brackets separate ladderons.

Although it is too early to conclude that Y does not come from total randomness, we can at least see that the whole string is well-organized (we may infer that EF is the most important in the sentence as it appears the most number of times, while MU, TEF, and BCDEF are also important). In fact, Y is converted from the sentence (with punctuations and capitals neglected): *my mother made a chocolate cake your mother made a chocolate cake my chocolate cake was delicious your chocolate cake was not delicious I ate half of my chocolate cake and you ate half* (denoting this sentence as Z for convenience). Every letter in Y corresponds to a word in Z. We can thus see that EF is *chocolate cake*, TEF is *my chocolate cake*, BCDEF is *mother made a chocolate cake* and MU is *ate half*.

There are four points needed to explain further. First of all, the ultimate reason that Y can be interpreted as Z is that Z is converted into Y in the first place on purpose. However, in fact, replacing *chocolate cake* by any noun phrase consisting of two words makes sense (as well as replacing *mother* by *father*, etc). So, in principle, Y does not contain any information about *chocolate cake*, but only the relationships among the parts of the sentence. That is to say, we can never interpret EF, BCDEF, or any single letter if there is no information from other places (this is also why some old languages are uninterpretable).

Secondly, for those substrings that appear more than once, they only contain non-redundant information once. For example, the first part of Z can definitely be compressed as *my and your* [*mother made a chocolate cake*] (although there might be grammar errors, the meaning is clear), that is, the information contained in this part [*mother made a chocolate cake*] as a whole does not change at all, but just with a different qualifier ahead.

The third point is significant but also confusing: the reason why we are able to interpret those non-repetitive substrings is that we wrote Z in the first place and then converted it to Y on purpose. If we only look at Y, there is no way we can tell if the letter that only appears once such as T and P is noise or contains useful information. Furthermore, after we replaced T by *my*, why we can interpret my as the actual meaning “of mine” is because the word my has already been repeated in my memory. A person who does not speak English does not have the word my repeated in his/her memory, so even if you tell him/her the sentence Z, he/she cannot understand the meaning “of mine”. Therefore, fundamentally, if a letter, a string, a word, a phrase, etc. has meanings, or equivalently, contains information, it must repeat, either in your memory/mind, in the system (the sentence) itself, or somewhere else.

Fourthly, there is an important assumption in the interpretation above: We considered the single letter as the basic unit of information, i.e., the basic set of the ladderpath consists of single letters. In fact, we do not have sufficient reasons to do that, because although it is reasonable to consider repetitive single letters (e.g., R, H, and K) as the basic blocks, it may not make sense for non-repetitive single letters such as P and S. In fact, we could consider PS as one basic block since P and S never appear independently. We are used to separating P and S because they are separated in our own language. What if in the alien language, they as a whole are just one letter (from the signals we received, we do not have enough reasons to believe that P and S are two separated letters)?

So, defining the basic set is the first step, which is subjective. We can define the basic set either based on some assumptions, hypotheses or facts from other sources, or only based on the target block itself. For Y, we can define the basic set to be the set of individual letters as we already did above, yet we can also define it to be only the letters that are repeated. In the latter case, Y should be considered as $Y^{\prime }$ = T [BCD [EF]] R [BCD [EF]] [T [EF]] H K R [EF] H K [MU] [T [EF]] [MU], of which the shortest ladderpath is:$$J_{Y^{\prime }}=\left\{T\left(2\right),R\left(2\right),H\left(2\right),K\left(2\right),\mathrm{EF}\left(3\right),\mathrm{MU}\left(2\right),\mathrm{BCD}⫽\mathrm{TEF},\mathrm{BCDEF}\right\},$$
where BCD is considered to be one block since B, C and D always appear together. So, the length of $J_{Y^{\prime }}$, also the ladderpath-index of $Y^{\prime }$, is $\lambda (Y^{\prime })=18-1-1=16$
*lifts* where 18 is the size-index of $Y^{\prime }$, the first 1 is the size-index of TEF (since EF is the basic block, we only need 1 *lift* to generate TEF), and the second 1 is the size-index of BCDEF (since BCD and EF are both basic blocks, we only need 1 *lift* to generate BCDEF). So, we can see that the ladderpath-index is different if the basic set is different (note that as long as the basic set is defined, it should not be changed in the following analyses). As for what basic set we should define, it is a different game. It should depend on what specific questions we are asking, which leads to the discussions in the next subsection.

### 4.2. On Information: Human Language (Non-Isolated/United System)

Imagine we put the sentence Z (first mentioned in Section 4.1) in a human language (English in this case). Then what basic blocks shall we choose: single letters, single words, or only those repeated? The most straightforward answer is single words, because the single word is the basic unit that makes any sentence. In fact, there is no problem to define the basic block to be the single letter, but it is just more complicated, since there will be an extra level in the ladderpath, i.e., the level that makes words from letters. Yet in principle, the two ways have no real difference, but like measuring the same quantity in a different unit.

However, in the framework of English, it is inappropriate to define the basic set to be the set of only those repeated in the sentence, because although some word does not repeat in Z (e.g., not), it repeats in the framework of English, that is, it can appear in other possible English sentences. Therefore, when we discuss questions under the framework of English, we have actually considered the target sentence and all other possible English sentences as a whole (namely, a ***non-isolated /united system***) where even though a word did not repeat in the target sentence, it repeats in this non-isolated system, and we thus must consider these words as basic blocks.

The alien signal Y in Section 4.1 is an example of an isolated system, i.e., there are no other signals or sentences that can be considered together with Y to be a united/non-isolated system. Thus, the letters that are not repetitive in Y are not repetitive in the whole isolated system either (namely, the sentence itself), which is why we have no strong reason to consider these non-repetitive letters as basic blocks. Therefore, for the isolated case, we have two options, both mentioned in Section 4.1: After making some assumptions, take single letters as basic blocks as in $J_{Y}$; Or only take repetitive letters as basic blocks and consider non-repetitive ones as noise, as in $J_{Y^{\prime }}$. Both approaches are correct, which are two interpretations of the original sequence. As for which approach reflects more of the reality, it is a totally different question, which should be addressed from other perspectives (referring to Appendix E for how we can apply this isolated/united-system idea to look for the evidence of intelligent life).

There is another point associated with the example in Section 4.1 that needs to be addressed: if we received many other signals of the same type as the signal Y, then what should we do? In this case, we have to consider Y and all of the other signals we have received as a united system, then we need to analyze the ladderpath of this entire united system (whose size-index is evidently very high). If at the end we found that only BCDEF repeats once, then the calculated ladderpath-index would be very close to the size-index and the order-index would be very low, suggesting that the system is more likely to be random. Furthermore, what if there is a virtually infinite number of signals in principle but we only received one of them, which happened to be the Y? In this case, we have to accept the fact that we can only consider Y alone as an isolated system, which may lead to a wrong or over-extrapolated conclusion (so extra evidence may be necessary, but that is an experimental problem outside of current considerations).

We can now come back to the question put forward in Section 1: We see that the “information” of a sentence in a language actually contains two levels of meanings. The first is the information in the narrow sense, which refers to the “information” we can extract from the united system consisting of the sentence itself and the language it belongs to, namely, the ladderpath. The second level of information is above the first level, referring to what reality, real object, behavior, etc that each ladderon corresponds to. The first level is about the sentence itself (namely the so-called syntactic information), while the second level is about the linkage between the sentence and the reality (namely the so-called semantic information).

### 4.3. On Origins of Life

As for how small the probability of the origin of life is, there is a famous metaphor called “junkyard tornado” [38]: The chance of emergence of life would be comparable to the chance that a tornado sweeping through a junkyard assembles a Boeing airplane. This is a vivid metaphor that is meant to help us comprehend how unlikely the emergence of life is. Yet it is misleading. The biggest flaw of this metaphor is that it suggests that the difficulty of making an airplane lies in assembling it with the most basic units (e.g., screws, semiconductors, or small elements lying in the junkyard), which is, however, not correct. Here we clarify this flaw.

How difficult is it for a person to make an airplane? First of all, let us simplify this question as much as possible: We assume that the airplane is just made of four parts, i.e., engine, propellers, wings, and control circuit.

A person living 10,000 years ago, would need to invent the four parts first, and then assemble them together. So the difficulty for them to make the airplane is the sum of the difficulty of inventing the four parts plus the difficulty of assembling. However, inventing all of the four parts altogether is easier than inventing each of them independently. For example, they need to invent metallurgy to obtain metals, but they only need to invent metallurgy once although all of the four parts require metals. The same reasoning applies to aerodynamics and wings and the propeller. There are also two wings, two propellers, and two engines, but he only needs to invent each of these parts once.

Therefore, the difficulty to make an airplane is not the sum of the difficulties of making each part individually, but that with repetitive parts excluded. This is the first level of meanings that the statement “making airplane is simpler than expected” implies.

The second level of meaning is that the basic set is different, that is, if the person lives in the 20th century all of the four parts are already matured technologies, but not the case for the ancients. Under the concept of ladderpath, for a modern person, the basic set consists of the engine, the propeller, the wing, and the control circuit, so an airplane’s ladderpath-index (also corresponding to the difficulty) is just four *lifts*. However, for the ancients, the basic blocks are merely ore and petroleum. Thus, the difficulty of making an airplane is much larger than 4 *lifts* in this case.

That is to say, in the 20th century, it is relatively easy for humans to invent airplanes, i.e., the probability of the emergence of airplanes is relatively large in the 20th century; while in ancient times, the probability of the emergence of airplanes is very small. The basic set the modern people face is the set derived from the set the ancient faces that has already undergone lots of generation-operations and generated lots of ladderons of higher levels.

The metaphor above about life actually compares life to the airplane, and compares atoms in the physical world to the scrap parts lying in the junkyard. Indeed, the probability that numerous atoms are assembled into a living system such as a cell by a simple physical process such as a tornado is extremely small. However, on one hand, life has a relatively low ladderpath-index (compared to its size-index) since it has many repeating elements (as we shall discuss below). On the other hand, life did not emerge all of a sudden from non-living systems, but probably through an elaborate path that generates ladderons, which gradually makes the system more and more complex, resulting in life at the end, like the “emergence” of the airplane in the 20th century.

Here we discuss the first point specifically: life has a relatively low ladderpath-index (compared to its size-index), due to its repetitive parts. First of all, ladderpath applies to any kind of object, including molecules. As for how to define the generation-operation and the length unit of ladderpath for molecules, it is different from the cases of strings. Here I give a reasonable scheme (there could be other schemes, see [33] for another attempt where molecules are considered to be composed of bonds, rather than atoms):The generation-operation for molecules is defined as follows: combine several molecular structures or fragments (namely, the blocks) into one, whereas the combination means that chemical bonds between atoms are formed;The length unit of a ladderpath is defined to be one chemical bond formed. That is, if *n* chemical bonds are formed in one generation-operation, the length of this generation-operation is *n* *lifts*.

Note that there is a range of possible ways in which molecular fragments could be combined. So, generation-operations could have different types (as we have mentioned after the definition of generation-operation in Section 2.2). For example, CH${}_{3}$CHCH${}_{2}$, CH${}_{3}$ and H could be combined into CH${}_{3}$CH${}_{2}$CH${}_{2}$CH${}_{3}$ (namely, butane) or HC(CH${}_{3}$)${}_{3}$ (namely, isobutane). The choice of generation-operations should be recorded along with the calculated ladderpath as the necessary context.

For lots of molecules, the difficulties of making them (i.e., one aspect of complexity that is described by the ladderpath-index) are smaller than expected (“expected” refers to “assembling individual atoms to make a molecule”). For example, NADH has two ribose groups, so NADH’s ladderpath-index is smaller than the number of atoms it contains; The backbone of RNA is made of lots of identical ribose rings and phosphate groups, and many types of proteins (such as tetrameric proteins) are made of repetitive structures, so their ladderpath-indices are smaller than the number of constituted atoms; Gene sequences have many repetitive segments, so their ladderpath-index is also smaller than the number of constituted base pairs. Furthermore, if we consider a group of molecules altogether (i.e., in the case of a target system, instead of an individual target block), the ladderpath-index will be smaller than the sum of individual ladderpath-indices. For example, if we consider 1 *mol* of NADH, 1 *mol* of ATP and 1 *mol* of FAD together, lots of their structures are repetitive.

We often say a protein is complex because we are wondering that although it contains so many atoms, why it is still so “ingeniously designed” [39]. However, after we extract the repetitive structures of proteins, its “complexity” is much smaller (more precisely, its ladderpath-index is relatively small, compared to its size-index). The same argument applies to life. As we consider a living system (e.g., a cell) as a whole, its complexity is much smaller than the sum of the complexities of each individual part. Although it could still be very complex, it makes the emergence of life look easier.

### 4.4. On Why Life Is Ordered

Let us begin with an observation. For a ladderpath-system (namely, any system that satisfies “being able to generate new blocks” and “replication”), each time a new ladderon is generated, the ladderpath-index of the whole system is increased (equivalently, the information the whole system contains is increased); each time a ladderon is replicated, the order-index of the whole system is increased. Therefore, the ladderpath-index and the order-index of a ladderpath-system increase naturally, i.e., a ladderpath-system naturally evolves towards “complex”, which is an inevitable consequence of its two essential properties: “being able to generate new blocks” and “replication”.

Now, before we ask *why life is ordered*, let us first ask: How do we interpret life is ordered, equivalently, well-organized? It is an intuitive thesis easily obtained from ordinary observations. We have two ways to interpret “ordered” here. Taking a cell as an example, the first interpretation is straightforward: it means a vast number of atoms organized into large biomolecules, which are organized into organelles, and in turn into a cell.

The second interpretation is that the atoms constituting the cell are unable to spread evenly in its space, and they are restricted at the positions, energy-levels and states of the large biomolecules and organelles (which is an interpretation in the sense of thermodynamics). As a result, the total number of all possible states in the state space of the cell is much smaller than that of an ideal gas (a typical example in thermodynamics) of the similar size. This implies that the cell’s thermodynamic entropy is low (referring to Appendix F for the discussion on the connections between the ladderpath and Shannon entropy).

The first interpretation refers to “ordered in the sense of the ladderpath”, while the second interpretation clearly refers to “ordered in the sense of thermodynamic entropy”. We can see that the two interpretations of “ordered” are not equivalent, although either makes sense in its own perspective.

Depending on how we interpret “ordered” we will approach the question *Why is life ordered?* very differently. Under the concept of ladderpath, this question might be solved readily, because evolving towards more and more ordered states is an intrinsic property of a ladderpath-system (to which life belongs), determined by the underlying evolution mechanism as discussed before. Yet, under the framework of thermodynamic entropy, this question is still a bit mysterious, because according to the second law of thermodynamics the entropy of a closed system spontaneously increases. Although, as an open system, the decrease of life’s entropy does not violate the second law, we still do not understand why/how life organizes itself so that it can resist the increase of thermodynamic entropy (some authors used more general/vague but more intriguing statements [40,41], e.g., “the entropy of life tends to decrease”, “life feeds on negative entropy”, etc., but all of these statements are basically equivalent).

If the following two statements are true, that is, “life must be a ladderpath-system (or put differently, life organizes itself into a ladderpath-system)” and “a ladderpath system is able to resist the increase of thermodynamic entropy”, then the question “why/how life organizes itself so that it can resist the increase of thermodynamic entropy” is solved. While it has been demonstrated above that life is a ladderpath-system, we thus need to figure out whether a ladderpath-system is able to resist the increase of thermodynamic entropy. The answer is likely to be yes, but a full proof/demonstration could not be provided here, as it requires much more work, both theoretically and empirically. Nonetheless, at least this type of question such as *Why is life ordered?* has been pinned down to a better understanding of ladderpath-systems. Since ladderpath-systems have a rigid mathematical definition, these questions are easier to pursue. By these discussions, we hope to convey the idea that ladderpath can provide a different angle when investigating this type of questions such as why life is ordered.

## 5. Conclusions

In this paper we have argued for the need for a new method for characterizing the complexity of objects that are neither drawn from a statistical distribution nor infinitely large (e.g., infinite strings). We have presented the ladderpath approach which decomposes a single finite object into a partially ordered multiset which describes how the object can be formed by joining or altering existing building blocks. From the minimal ladderpath we can compute two measures of information, namely the ladderpath-index and the order-index, which capture two distinct dimensions of complexity: the difficulty of generating the object and the degree of order. This framework forces a distinction between isolated and united systems, and also puts our understanding of the origin of life and evolution into a clearer view. Here we only scratch the surface of many possible areas of application and it is our hope that this approach will be successfully applied in fields such as linguistics, technological evolution, and exobiology. 

## Figures and Tables

**Figure 1 entropy-24-01082-f001:**
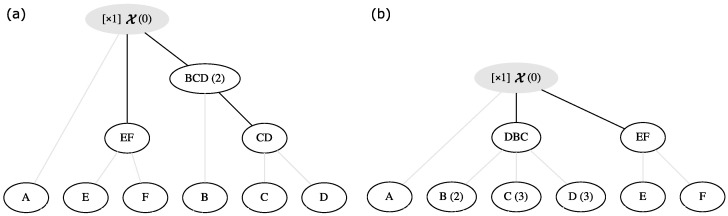
(**a**) The laddergraph that corresponds to ladderpath $J_{X,1}$. (**b**) The laddergraph that corresponds to ladderpath $J_{X,2}$. Grey blocks represent target blocks. “[×1]” in front of the target block represents that “we need to obtain 1 of such target block in the end”. If (n) is added behind a block, it means the multiplicity (in the partially ordered multiset representation) of this block is *n*, while if there is no (n) behind, it means the multiplicity is 1. Finally, (0) behind the target block means that its multiplicity in this ladderpath is 0 (in principle, we should not draw blocks if their multiplicities are 0’s, but here we explicitly drew the target block, just in order to show the readers the hierarchical relationships between it and other blocks).

**Figure 2 entropy-24-01082-f002:**
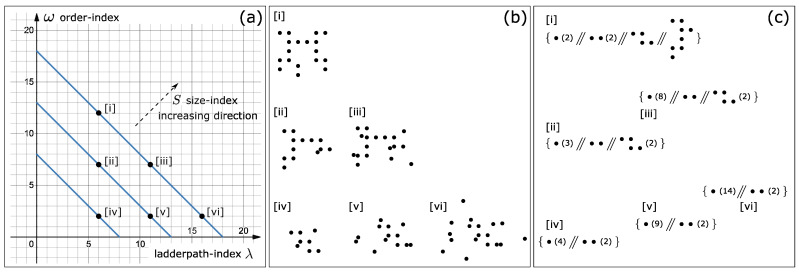
(**a**) The relationships among the ladderpath-index λ, the order-index ω and the size-index *S*. The blue diagonals are the contour lines of *S*. The points [i]–[vi] correspond to the patterns in (**b**) (note that one coordinate could correspond to an infinite number of patterns), and the coordinates are (6,12),(6,7),(11,7),(6,2),(11,2),(16,2), respectively. (**b**) The patterns corresponding to the six coordinates in (**a**). (**c**) The ladderpaths of the six patterns.

**Figure 3 entropy-24-01082-f003:**
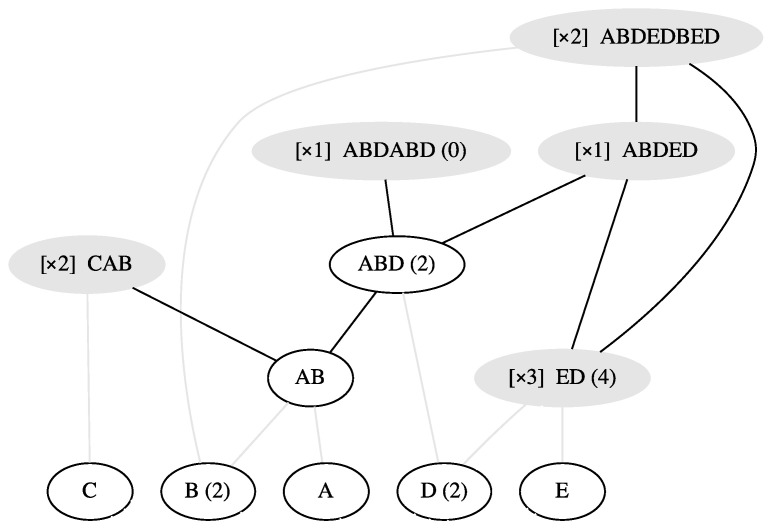
The laddergraph representation of one ladderpath of the target system Q (this ladderpath $J_{Q}$ is actually the shortest one for Q). All of the grey blocks constitute the target system Q. “[×n]” in front of the grey blocks represents that there are *n* such blocks included in the target system Q. If (n) is added behind a block, it means the multiplicity (in the partially ordered multiset representation) of this block is *n*, while if there is no (n) behind, it means the multiplicity is 1. Finally, (0) means that its multiplicity in this ladderpath is 0 (in principle, we should not draw blocks if their multiplicities are 0’s, but here we explicitly drew them, just in order to show the readers the hierarchical relationships among important blocks).

## Data Availability

There is no data for this paper. The code is available at https://github.com/yuernestliu/ladderpath (accessed on 2 August 2022).

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
