# Peer review of "Ladderpath Approach: How Tinkering and Reuse Increase Complexity and Information"

_entropy, 2022, doi:10.3390/e24081082_

Round 1

Reviewer 1 Report

I find this paper very interesting, and I find the final discussion about the origin of life based on the remark that repetition makes easier the invention of life very attractive.

However, having devoted a part of my research work on the use of Kolmogorov concept of complexity, I have doubts on the part of the paper devoted to the detection of extraterrestrial life. The authors use the example of the message

ACXLGICXGOXEMZBRCNKXACXLPICXEMZBRCNKX.

They notice that EMZBRCNKX appears twice. Which is the probability that a combination of 9 letters appears twice in a sequence 37 characters? Yes, I imagine that it is very small. However, which is the source of this message? If the message is only one, it is not difficult to guess that there is intelligence behind him. What about the case when there is a virtually infinite number of messages of the same length? In that case also this message may be the result of randomness. I invite the authors to make comments on this issue.

The authors conclude their manuscript stressing that much more work is necessary to establish a connection between their tinkering approach and the widely accepted conviction that life involve negentropy. I like these comments.

I am inclined to recommend this paper for publication with the invitation to make some improvements. In addition to the above illustrated doubt, I have to say that I find not easy to follow their Sections 2.2 and 2.3. However, I imagine that that the readers interested in using their method will be able to apply their prescription.

Reviewer 2 Report

In the abstract, there is an error in the syntax of the following sentence:

Many measures of these quantities, either rely on a statistical notion of information, are difficult to compute, or can only be applied to strings.

the alternative "either ... or" is complicated by the insertion of a superfluous colon : "are difficult to compute"

The more I think about this article, the more I think that it grasps too much.   The sections 4.3. and 4.4. dedicated to the complexity of life seem superfluous. I would suggest renouncing altogether the parts that delve into the complexity of life and to recenter the whole structure of the article to computational linguistics.

Reviewer 3 Report

  1. I read the paper with real interest. It proposes some new ideas, which seem to be fresh, but at the end it brings a bunch of provocative statements, which actually do not push the science ahead very far.

    Detailed remarks include:

    1. You state, that the idea is novel and that it is substantially different from all that was published above. I rather feel, that the idea of Lempel and Ziv Algorithmic Complexity, being „the code of a shortest algorithm, which can build the analyzed sequence” is actually quite close to the idea of ladder path creation.
    2. In my opinion, in the introduction you omit various aspects of complexity and discuss only the ones that are relevant from the point of view of your theory. One specific example is self-similarity and scaling. They can be obtained very easily, which may be observed e.g. on Koch snowflake, the constructal law of which is trivial. Would you describe it as complex or ordered? Still if you use stochastic rules, like in IFS, you may obtain quite regular structures - and you seem to contradict randomness and order. There are many types of constructal laws, that induce self-similarity, including Murray Law, so the idea comes directly from the nature. However the introduction is clearly the best part of the paper, and the collection of papers cited there is interesting as such.
    3. Also, you don’t refer to statistical vs sequential order. Shannon entropy is insensitive to sequence, it is only a popularity contest against basic symbols. Algorithmic complexity or dictionary compression algorithms are, in turn, sensitive to sequence. Having written as much on complexity, you should consider adding this classification.
    4. Negative multiplicity is not natural. Consider the change.
    5. My first questions at line 202/203 was, is the order in sequence DDFAC important or not, but soon I realized that it is much more. In mathematical terms, I would say, that ladderpath, order and size index are under constraint, which is the actual build set. The build set already contains certain information, which is either known a priori (as the alphabet) or derived from the data, which makes this build set case-specific. So there is a clear relation between the ladder path indices and the complexity of the build set itself, and this issue is completely left behind. Also the frequency of occurrence of certain building blocks is not discussed, and this would lead the measures closer to Shannon. The indices you introduce are not sensitive to the popularity of building blocks that are used in the ladderpath.
    6. If you extrapolate the relation that less ordered sequence carries more information, you get to a contradiction, that the most information is carried by white noise of no structure. And this is exactly what you tried to avoid (in introduction). Therefore, the theoretical framework, which has been developed, is not consistent with the assumptions.
    7. In 423-431 I started to think of alphabet of alphabets. The idea of recoding is absent, however e.g. in proteins it is obvious: nobody describes NA or Ach by its genetic code.On different levels of organization we have different alphabets
    8. In 591-594 you mention analysis of digraphs, which has its long tradition in cryptography - i.e. exactly where the work of Shannon is rooted.
    9. 611 the „qualifier ahead” sets the context of the information, and the context changes with the communicate. If I start the sentence with: don’t listen to the next statement, as it does not carry information, then the information related to this statement is irrelevant. There is an interesting book by Lind and Marcus, An Introduction to Symbolic Dynamics and Coding, which i.a. covers this topic. 
    10. The weakest part is the discussion, and in my opinion it fails to demonstrate the scientific quality of the method and convince us to use it. Moreover, the important relation between the indices and the basic set is not disclosed in the definition of the measure. Its only merit seems to be the opening possibility to coin journalistic statements as: „Life is not as complex as expected”. Really? Concerning DNA (740) there is an interesting example, that there exist non-coding parts, so the information may be judged only ex post by observation which proteins are being synthesized and the apparent complexity of non-coding parts is not important at all
    11. 750 I cannot agree that the complexity of the system is much smaller than the complexities of each individual part. How about emergent phenomena? Coordination, synchronization etc, they are important everywhere, from life science to modern warfare.
    12. 763: if the molecules are placed randomly, any pattern is legitimate. Likewise, nine, nine, nine is a legitimate result of random number generator (Credits to Dilibert)
    13. When you mention the patterns that appear when ladders get in relation, you forget, that this relation may obtain several distinct forms, depending on environmental conditions. The „+” sign shows itself as unique, which it isn’t.
    14. 770 When you describe degrees of freedom, you mention quantum, but the majority of states you refer to are classical. Also the description of micro state is rather vague. Also such terms as state space or number of degrees of freedom will be helpful. 
    15. Consideration on Second Law of Thermodynamics are imprecise. The fact, that der Entropie der Welt streibt ein Maximum zu does not rule out the possibility, that in certain area of the state space the entropy actually decreases.
    16. The paper made me think. Are small proteins more popular than large? Is the protein popularity contest won by the shortest sequences?
    17.  

Round 2

Reviewer 2 Report

It is OK   You can publish

Reviewer 3 Report

No further comments. It was interesting to discuss.

This manuscript is a resubmission of an earlier submission. The following is a list of the peer review reports and author responses from that submission.

Round 1

Reviewer 1 Report

The authors propose a new method for characterizing the complexity of objects. Objects subjected to this method cannot be derived from the probability distribution and cannot be infinite.

This concept is based on decomposition the object into a partially ordered multiset which describes how the object can be formed by joining or altering existing building blocks.

This article is not easy to read. However, I would not take this as an objection. It requires increased attention. Moreover, the authors made sure that the reader had the best possible ease. It's a great idea to create a glossary of terms.

In my opinion, the article is worth publishing. I would only recommend developing an introduction. I was surprised that there was no mention of the importance of Fisher's information, both in the study of the complexity of the objects and in the general description of the information it carries.

Of course, the authors do not use it in their model. However, it is worth mentioning because it is closely related to the model in question. A number of papers on the relationship between Fisher Information and the measurement of complexity are still published, see e.g.

https://doi.org/10.1093/imaiai/iaaa033

or information carried by market parameters:

https://doi.org/10.3390/e23111464

Note that the proposed approach can also be used in the analysis of signals generated by the market. They form certain regularities that allow us to predict (for better or worse) the behavior of the markets in the future. It is therefore worth mentioning the importance of Fisher's information. This will give the reader a more complete picture of the potential applications of the proposed method. This is especially important when proposing something new. As we know, it is very difficult to break through with new topics.

Reviewer 2 Report

\documentclass[12pt]{article}
\begin{document}
\section*{
Referee report on
``Ladderpath Theory: A New Look at Complexity and Information'' by
Yu Liu , Zengru Di and Philip Gerlee
}
The authors initially present a way recursively to identify repetitive components in long strings, and show how to use this knowledge to 
compress the string, that is, to map the original string onto a shorter one, from which the contents of the initial string
can be fully recovered. The authors argue that this can be used for structures more general than strings, but the way in which they perform 
such extensions is the straightforward mapping of these more general (say 2-dimensional) structures, on strings. The 
possible amount of compression can then be used to provide a measure of the information contained in the string. They then proceed
to considerations of a more general nature concerning the origin of life and other matters of general interest. 

The general considerations which end the paper appear to be of significant interest, but there is, in my view, an earlier issue which 
must be addressed: essentially the ``ladderpath'' algorithm described by the authors is nothing else than a variation on the LZ compression
algorithm proposed in 1978 by A.~Lempel and J.~Ziv. The authors dismiss this with the remark: ``Zempel-Liv is related to the optimal rate 
of lossless compression and does not account for the hierarchical structure of a sequence''. This is questionable on two counts: first, 
the ladderpath algorithm developed by the authors also is of the lossless compression type, and second, it is simply wrong to state
that the LZ algorithm does not take into account the hierarchical structure possibly present in the input. Indeed, the ``ladderpath'' algorithm
is best viewed as some variant on LZ. There are technical differences, but these refer mainly to the fact that the LZ algorithm is fully
specified, whereas the ``ladderpath'' algorithm is not. 

The authors now claim a relation between Shannon entropy for an isolated sequence and ladderpath compression. Such remarks are undoubtedly
correct, but they do not appear to be new: the articles quoted at the end of this report all have something to say on this subject. 

Finally, it is not clear to me to what extent the claim made in the Abstract is fulfilled in the paper: ``From the ladderpath two measures 
naturally emerge: the ladderpath-index and the order-index, which represent two axes of complexity.'' But it appears that these two measures
simply add up to the total {\em length\/} of the string: surely, the 2 are thus not in any meaningful 
sense independent measures of complexity. The paper thus does not appear to yield anything else than the variation on the Shannon entropy
for individual signals already discussed in the literature. 

As it stands, the paper does not adequately describe considerable previous work on this and related subjects. Further, it is not clear to this author that
there is anything significantly new in the paper, which should therefore  be rejected.

\begin{enumerate}

\item Jacob Ziv, IEEE Transactions on Information Theory,  Vol.~IT-24, No.~4, JULY 1978, p.~405,
Coding Theorems for Individual Sequences

\item Jacob Ziv and Abraham Lempel, IEEE Transactions on Information Theory,  Vol.~IT-23, 
No.~3, MAY 1977, p.~337
A Universal Algorithm for Sequential Data Compression

\item Jacob Ziv, and Neri Merhav, IEEE Transactions on Information Theory,  Vol.~39, No.~4, JULY 1993, p.~1270
A Measure of Relative Entropy
Between Individual Sequences with
Application to Universal Classification

\item Jacob Ziv and Abraham Lempel, IEEE Transactions on Information Theory, Vol.~IT-24, No.~5, September 1978,
p.~530, Compression of lndividual Sequences via
Variable-Rate Coding

\item Hansel G., Perrin D., Simon I. (1992) Compression and entropy. In: Finkel A., Jantzen M. (eds) STACS 92. STACS 1992. Lecture Notes in Computer Science, vol 577. Springer, Berlin, Heidelberg.\\ https://doi.org/10.1007/3-540-55210-3\_209
\end{enumerate}

\end{document}

Round 2

Reviewer 2 Report

\documentclass[12pt]{article}
\begin{document}
\section*{
Referee report on
``Ladderpath Theory: A New Look at Complexity and Information'' by
Yu Liu , Zengru Di and Philip Gerlee
}

In the authors' reply, as well as in the article's text, the claim that the results are new is repeated in a way that makes 
it straightforward to show it to be questionable. In the Abstract we have the following 2 sentences: ``Some measures 
of these quantities, 
such as Shannon entropy and related complexity measures, are defined for objects drawn from a 
statistical ensemble and cannot be computed for single objects; While some other measures such as 
Kolmogorov complexity suffers from incomputability. Based on assembly theory, we attempt to fill
this gap by introducing the notion of a ladderpath which describes how an object can be decomposed 
into a hierarchical structure using repetitive elements.'' as well as ``The ladderpath theory provides a novel 
characterization of the information that is contained in a single object (or a system)''.

What I wish to dispute is both the existence of a ``gap'' as argued in the first sentence, as well as the fact that
the  ladderpath theory provides a ``novel'' characterization of the information contained in a single system. 

Indeed, the very titles of the references

Abraham Lempel and Jacob Ziv,
On the Complexity of Finite Sequences, IEEE Transactions on Information Theory,
 IT-22, NO. 1, January 1976  pp.~75--81

Jacob Ziv, IEEE Transactions on Information Theory,  Vol.~IT-24, No.~4, July 1978, p.~405,
Coding Theorems for Individual Sequences

\noindent show that an information measure can be defined for an individual sequence, and that it is estimated, as follows 
from the theorems stated in the paper, by the use of the LZ compression technique as well as related ones. Furthermore, 
the complexity measure defined in the first of the above references is in fact extremely close to the definition
given in the authors' paper, as it also involves the estimation of a minimal procedure to build up the sequence
out of its building blocks. 

True, in order to prove convergence theorems, one needs infinite sequences, but there is no difficulty of principle in applying 
these same techniques to large but finite sequences. To be specific, the information measure which is estimated
is defined as follows: let $2^{lh_l(u)}$ be the total number of words that appear as you slide a window of size $l$
along the infinite sequence $u$. It is seen that $\lim_{l\to\infty}h_l(u)=h(u)$ exists, and that its value can be estimated
by the LZ compression approach. 

Note that the above publications are 35 years old. It is of course understandable that such work, which has not made its
way into the textbooks, could be missed. However, ignoring the existence of previous work when rediscovering it,
is not the appropriate approach. 

Finally, the answer made to my criticism, that the claim of having ``two axes of complexity'' is in fact illusory, since
$\lambda+\omega=S$, where $S$ is the signal length, the authors answer as follows: ``It is true indeed, but in fact, any target
can be placed in a particular position in these coordinates. This is because although the three indices are
constrained by $\lambda + \omega = S$, two of them are free.'' Fair enough, but when the authors speak of 3 indices,
do they really mean that $S$, the size of the message, is actually an ``index of complexity''? Surely, it seems more
natural to normalize both $\lambda$ and $\omega$ by $S$, which leads to the normalized values adding up to one. 
That 2 values which add up to one do not form ``two axes of complexity'' should be clear. 

In other words, I remain wholly unconvinced by the authors' arguments and by their answers to my remarks, and
stand by my earlier opinion. The paper should be rejected.
\end{document}

Round 3

Reviewer 2 Report

\documentclass[12pt]{article}
\begin{document}
\section*{
Referee report on
``Ladderpath Theory: A New Look at Complexity and Information'' by
Yu Liu , Zengru Di and Philip Gerlee
}
The central issue of my earlier referee report was that the method developed in Section 2 to analyse the complexity 
of individual sequences is not original. The LZ compression method was used exactly to this end in the late seventies.
That the two methods are not rigorously identical does not meet the criticism: presenting a mild variant of LZ compression
and using it to the same purpose of complexity analysis for individual sequences does not give a new result. The result might
be of interest if the new approach were to yield qualitatively new results, compared to the old method. This does not seem 
likely, however, and the authors do not engage in any such comparisons. The authors add that their approach allows to 
compress structures somewhat more general than strings, such as two-dimensional dot patterns. It would be necessary to look
at the literature to find out whether such work has already been done. In any case, however, the extension to dot patterns is
not systematically explained, nor does it, in my view, represent a sufficient extension of the LZ algorithm to warrant publication 
on its own. In any case, however, the article spends a considerable amount of time on an algorithm which should be treated
altogether by an explicit reference to LZ, saying that the entire approach championed in Section 2 was described there. 
This would mean a significant reorganisation of the paper, which the authors do not undertake. 

Finally, concerning the ``two axes of complexity'', I still find the authors' views more than a bit puzzling. They say for instance
on p.~11:
\begin{quote}
 It deserves to mention that we may intuitively say that [vi] is more  ``complex'' than
 [i], because the former looks more random/irregular/difficult to reproduce; while we
 may also say that [i] is more  ``complex'' than [vi], because [i] needs more detailed, complex
 and delicate mechanisms to generate. However, it is not difficult to realize that these two
 ``complex'' refer to two distinct directions\ldots
 In fact, the two directions correspond to the ladderpath-index $\lambda$  and the order-index $\omega$ ,
 respectively. Therefore, now we are able to distinguish the two axes of ``complexity''.
\end{quote}
But [i] and [vi] have the same size, so that $\lambda$ and $\omega$ add up to the same quantity. In other words, any
increase in $\lambda$ yields an equal decrease in $\omega$, and viceversa. In other words, complexity along the ``first axis''
is exactly the contrary of complexity along the ``second axis''. Simply said, the two axes point in exactly opposite directions. 
``Higher complexity'' in the first sense of the word is exactly equivalent to ``lower complexity'' in the second sense of the word.
This is hardly a terminology calculated to bring clarity to a difficult subject. And finally, I still find it bizarre to compare the complexity
of two signals with different sizes, as well as to think of size as being an ``axis of complexity''. 

In my view, the authors have altogether failed to address the core of my remarks which simply states that the bulk of the paper
reports results which are not original.  The paper should be rejected.
\end{document}